# Geometric Mean Serum Cotinine Concentrations Confirm a Continued Decline in Secondhand Smoke Exposure among U.S. Nonsmokers—NHANES 2003 to 2018

**DOI:** 10.3390/ijerph19105862

**Published:** 2022-05-11

**Authors:** Kevin T. Caron, Wanzhe Zhu, John T. Bernert, Lanqing Wang, Benjamin C. Blount, Kristin Dortch, Ronald E. Hunter, Tia Harmon, J. Ricky Akins, James Tsai, David M. Homa, James L. Pirkle, Connie S. Sosnoff

**Affiliations:** 1Division of Laboratory Sciences, Centers for Disease Control and Prevention, Atlanta, GA 30341, USA; poq5@cdc.gov (W.Z.); jtb2@cdc.gov (J.T.B.); lfw3@cdc.gov (L.W.); bkb3@cdc.gov (B.C.B.); iop2@cdc.gov (K.D.); ronald.e.hunter@gmail.com (R.E.H.); harmon.tia@gmail.com (T.H.); rickyakins@netzero.net (J.R.A.); jlp1@cdc.gov (J.L.P.); css3@cdc.gov (C.S.S.); 2Office on Smoking and Health, Centers for Disease Control and Prevention, Atlanta, GA 30341, USA; jxt9@cdc.gov (J.T.); dgh3@cdc.gov (D.M.H.)

**Keywords:** secondhand smoke, biomarker, serum cotinine, tobacco exposure, NHANES, nicotine

## Abstract

The objective of this study was to examine long-term trends in serum cotinine (COT) concentrations, as a measure of secondhand smoke (SHS) exposure, in U.S. nonsmokers using data from the National Health and Nutrition Examination Surveys (NHANES) from 2003 to 2018. We analyzed NHANES serum COT results from 8 continuous NHANES 2 year cycles from 2003 to 2018 using a liquid chromatography–tandem mass spectrometry assay that has been maintained continuously at the Centers for Disease Control and Prevention (CDC) since 1992. Serum COT concentrations (based on the geometric means) among nonsmokers in the U.S. decreased by an average of 11.0% (95% confidence interval (CI) [8.8%, 13.1%]; *p* < 0.0001) every 2 year cycle. From 2003 to 2018, serum COT concentrations in U.S. nonsmokers declined by 55.0%, from 0.065 ng/mL in 2003–2004 to 0.029 ng/mL in 2017–2018 (*p* < 0.0001). Significant decreases in serum COT concentrations were observed in all demographic groups. While disparities between these groups seems to be shrinking over time, several previously observed disparities in SHS exposure remain in 2017–2018. Serum COT concentrations of the non-Hispanic Black population remained higher than those of non-Hispanic Whites and Mexican Americans (*p* < 0.0001). Additionally, serum COT concentrations were significantly higher for children aged 3–5 years than other age groups (*p* ≤ 0.0002), and men continued to have significantly higher serum COT concentrations than women (*p* = 0.0384). While there is no safe level of exposure to SHS, the decrease in serum COT concentrations in the U.S. population as well as across demographic groupings represents a positive public health outcome and supports the importance of comprehensive smoke-free laws and policies for workplaces, public places, homes, and vehicles to protect nonsmokers from SHS exposure.

## 1. Introduction

In 1964, the U.S. Surgeon General released their first report on smoking and health, concluding that cigarette smoking is a health hazard of sufficient importance to warrant corrective action and that smoking is a cause of lung cancer [1]. In a 2014 report outlining 50 years of progress, the Office of the Surgeon General affirmed a causal relationship between smoking and a number of chronic diseases and cancers including cancer of the liver, colon and rectum, lung, oral cavity and throat as well as coronary heart disease, chronic obstructive pulmonary disease and stroke [2]. While public health efforts have increased awareness of the risks associated with smoking and helped to bring about a gradual decline in smoking prevalence [3], tobacco smoking remains the leading cause of preventable death and disability in the United States [2,4].

Secondhand smoke (SHS) refers to the combination of sidestream smoke, which is smoke emitted into the environment from the smoldering tip of a combustible tobacco product, with smoke exhaled by the smoker [5]. SHS contains over 7000 chemicals, including hundreds that are hazardous and at least 69 known to cause cancer [6]. Additionally, reports from the National Cancer Institute, the U.S. Environmental Protection Agency, and the Office of the Surgeon General have established SHS exposure as a cause of many adverse health outcomes including lung cancer, coronary heart disease, and stroke [7,8,9]. The U.S. Surgeon General has concluded that there is no risk-free level of SHS exposure; even brief exposure can be harmful to health [2,10,11].

In 1990, an expert panel convened to assess the efficacy of existing biomarker measurements as an index of SHS exposure. The panel concluded that serum cotinine (COT) was the biomarker of choice for a quantitative assessment of tobacco exposure in nonsmokers [12]. A subsequent review by Benowitz et al. conducted in 1996 [13] as well as the 2006 Report of the Surgeon General [9] upheld serum COT’s status as the preferred biomarker for assessing SHS exposure. While the rise in use of electronic cigarettes (e-cigarettes) may contribute to increases in serum COT concentration among nonsmokers, it is unlikely to have a large effect primarily due to the absence of sidestream smoke as well as findings of minimal nicotine in exhaled breath of people who are established users of e-cigarettes [14].

COT is the noncarcinogenic, primary proximate metabolite of nicotine. It is specific for exposure to tobacco, and has a half-life of approximately 18 h, making it a preferred biomarker over nicotine, which has a much shorter half-life. While serum COT measurements only indicate tobacco exposure for individuals during the previous few days and are subject to interindividual variations in the metabolism of nicotine, these limitations prove inconsequential when comparing mean concentrations from populations [13]. SHS exposure may be assessed using the prevalence measure (percentage of nonsmokers with serum COT concentrations within a defined range) or geometric mean (GM) of serum COT concentrations. The first national estimate of exposure of the U.S. population to SHS based on measurements of serum COT concentrations was obtained from samples collected in the period 1988–1991 as part of the first phase of the Third National Health and Nutrition Examination Survey (NHANES III) [15]. A subsequent report used data from NHANES III and the next two 2 year NHANES cycles (1999–2000 and 2001–2002) to show that SHS exposure declined substantially among U.S. nonsmokers from 1988 to 2002 [16]. In this study, we have extended those findings over all subsequent NHANES intervals from 2003 to 2018 and documented that this significant decline in SHS exposure, as measured by the GM concentrations of serum COT, has persisted, demonstrating the success of continued public health efforts to minimize exposure in nonsmokers.

## 2. Materials and Methods

### 2.1. Overview

NHANES is designed to assess the health and nutritional status of the U.S. civilian, noninstitutionalized population. It is based upon a complex, stratified, multistage, probability cluster sample design (https://www.cdc.gov/nchs/nhanes/index.htm (accessed on 2 February 2022)). NHANES participants are interviewed at their place of residence before they visit a mobile examination center (MEC), where they receive a physical examination administered by health professionals. During the MEC exam, blood samples are drawn from each participant, coagulated, and centrifuged. Sample aliquots of serum are frozen and transported to the appropriate laboratory for analysis. Serum COT analysis is performed at the CDC. In 1999, NHANES became a continuous, cross-sectional survey conducted over the course of 2 year cycles. The data reported here represent 8 continuous 2 year survey cycles over a 16 year period, from 2003 to 2018. All NHANES studies were approved by the National Center for Health Statistics Ethics Review Board. All participants, or their parent or guardian, provided informed consent before participation in the surveys and documented assent was obtained for children and adolescents aged 7–17 years.

### 2.2. Participants

In this study, nonsmokers were defined as (1) children aged 3–11 years with serum COT concentrations ≤ 10 ng/mL; (2) adolescents aged 12–19 years with serum COT concentrations ≤ 10 ng/mL and who did not report smoking within the preceding 30 days and did not use any nicotine-containing products within the preceding 5 days; and (3) adults aged ≥ 20 years with serum COT concentrations ≤ 10 ng/mL and who did not report being a current smoker or use of any nicotine-containing product within the preceding 5 days. Participants self-reported sex (male or female) and race and Hispanic origin (non-Hispanic White, non-Hispanic Black, Mexican American, or other Hispanic/other race/multi-racial). Age was reported as the age in years at the time of the NHANES interview. This study included a total of 48,955 nonsmokers.

### 2.3. Cotinine Analysis

Serum COT concentrations were measured by an isotope-dilution, high-performance liquid chromatography/atmospheric pressure chemical ionization tandem mass spectrometric (ID HPLC-APCI MS/MS) method [17,18]. Serum samples were spiked with stable isotope-labeled COT as an internal standard. The sample was basified, and liquid–liquid extraction was used to extract COT from the serum matrix with methylene chloride. Then, the organic extract was concentrated, and the residue was injected onto a C18 HPLC analytical column for separation. The eluent from these injections was monitored by APCI-MS/MS. The mass-to-charge ratio of the quantification product ion (80) from the mass-to-charge ratio of the quasi-molecular ion (177) was used to monitor COT. Additional ions for the internal standard and for confirmation were also monitored during APCI-MS/MS analysis. COT concentrations were then derived from the area ratios of native-to-labeled compounds in the sample by comparison to a standard curve. All assays adhered to the rigorous quality control (QC)/quality assurance (QA) program maintained in the Division of Laboratory Sciences at the CDC [19].

### 2.4. Analysis Changes

The ID HPLC-APCI MS/MS method for detecting the concentration of COT in human serum has been maintained continuously in the same laboratory at the CDC since 1992, when it was first developed for analyzing samples from NHANES III. Since then, there have been changes to both the NHANES survey protocol and the laboratory method. These changes are documented in Appendix A Table A1. While the test principle of the serum COT assay has remained the same, advancements in sample extraction, method automation, and the incorporation of more sensitive instrumentation have helped lower the assay’s limit of detection (LOD) and decrease the volume of serum required for analysis.

Serum COT analyses of samples collected from 2003 to 2012 required the use of 0.5 mL of serum per analysis and utilized a Sciex API 4000 quadrupole mass spectrometer (Foster City, CA, USA) for quantification. COT was extracted by applying serum samples directly to basified extraction cartridges (Chem Elut columns, Varian, Harbor City, CA, USA) packed with diatomaceous earth. The LOD for samples analyzed using this method of analysis was 0.015 ng/mL [20].

NHANES samples collected from 2013 to 2018 were analyzed using an improved, automated sample cleanup system [21]. This method reduced the amount of serum required from 0.5 to 0.2 mL per analysis. Extraction of COT using the automated system was achieved using a 96-well extraction plate (Isolute SLE+ Supported Liquid Extraction Biotage, Uppsala, Sweden). Although the extraction principle and process for removing proteins remained essentially the same, the eluting solvent was changed to a mixture of 95% methylene chloride and 5% isopropanol to aid in the recovery of hydroxycotinine, a metabolite of COT, whose measurement was added to these analyses. Additionally, a newer, more sensitive mass spectrometer was used for quantification, the Sciex API 6500 (Foster City, CA, USA). The LOD for this method of analysis was maintained at 0.015 ng/mL [21].

### 2.5. Quality Assurance

Serum samples were measured in analytical runs that included aliquots from two distinct QC pools. These QC pools contained known analyte concentrations consistent with the serum COT concentrations of nonsmokers and people who smoke. QC pools were created from stock pools of human serum: a low stock pool from nonsmokers with minimum exposure to SHS, and a high stock pool from people who use tobacco products. These pools then were analyzed and combined to achieve target analyte concentrations [21]. The first QC pools for the COT assay were characterized in 1992 with the following concentrations: 0.268 ng/mL (95% confidence interval (CI) [0.208, 0.328]), 1.88 ng/mL (95% CI [1.64, 2.12]), and 207 ng/mL (95% CI [182, 232]) COT. All QC pools were homogenized, aliquoted into individual vials, and stored at −60 °C.

### 2.6. Long-Term Assay Stability

The COT assay has been maintained in a single laboratory since its inception, but it also has been improved on several occasions and fully automated since it was first implemented in 1992. Due to these ongoing changes made to improve the NHANES survey and serum COT method (Appendix A Table A1), further testing was implemented to assure the continuity, stability, and uniformity of measurements throughout the history of the assay. For example, since 1999, long-term method stability and precision have been monitored by periodically re-assaying aliquots of the original NHANES III QC pools prepared in 1992. This was feasible since COT in serum is stable indefinitely when stored at −60 °C or below.

### 2.7. Statistical Analysis

Data analyses were performed using SAS version 9.4 (SAS Institute Inc., Cary, NC, USA). Measurements below the LOD were replaced with LOD/√2. Statistical analyses were conducted using the NHANES examination sample weights as described in the NHANES analytical guidelines (https://wwwn.cdc.gov/nchs/nhanes/analyticguidelines.aspx (accessed on 2 February 2022)). Due to substantial right skew, serum COT concentrations were log transformed. To study changes in serum COT concentrations within the period 2003–2018 for U.S. nonsmokers exposed to SHS, we calculated sample-weighted GM serum COT concentrations by NHANES cycle and by race and Hispanic origin (non-Hispanic Black, Mexican American, non-Hispanic White, or other Hispanic/other race/multi-racial), age group (3–5, 6–11, 12–19, 20–39, 40–59, 60–85 years), and sex (males, females). Where applicable, we noted GM concentrations that did not meet the statistical reliability criterion of biomarker results greater than 60% above the LOD (Appendix A Table A2). Pairwise comparisons were conducted on GM serum COT concentrations across different survey cycles and demographics, and Bonferroni correction was used to adjust for multiple comparisons (Appendix A Table A3). To investigate the trend in serum COT concentrations from 2003–2004 to 2017–2018, we constructed a sample-weighted log-linear regression model, with serum COT concentration as the dependent variable and NHANES survey cycle as the independent variable. The average percent change and its 95% confidence interval (CI) were calculated per survey cycle by exponentiating the regression coefficients and their corresponding CIs. We also examined changes in serum COT concentration over time and across demographic groups via log-linear regression models, with NHANES survey cycle and either age group, sex, or race and Hispanic origin, and their respective interactions with NHANES survey cycle, as predictors (Appendix A Table A4, Table A5, Table A6 and Table A7). Statistical significance was set at α ≤ 0.05.

## 3. Results

The sample-weighted GMs and 95% CIs for serum COT concentrations in U.S. nonsmokers by NHANES cycle overall and by sex, race and Hispanic origin, and age are displayed in Table 1. Serum COT concentrations fell significantly (*p* ≤ 0.022) by at least 44%, and in some cases more than 60%, overall and for all demographic groups studied from 2003 to 2018 (Appendix A Table A3). Pairwise comparisons of sample-weighted COT GMs to describe disparities between nonsmoker demographic groups during the final survey period, 2017–2018, can be found in Table 2. Significantly higher serum COT concentrations were observed during the 2017–2018 survey in the non-Hispanic Black population as compared to Mexican Americans and non-Hispanic Whites (*p* < 0.0001). Additionally, children aged 3–5 years had significantly higher serum COT concentrations than older populations (*p* ≤ 0.0002), and the male population was more exposed to SHS than the female population (*p* = 0.0384).

The overall GMs and 95% CIs for serum COT concentrations in U.S. nonsmokers from 2003 to 2018 are depicted in Figure 1 along with the log-linear regression line. During this period, the GMs of U.S. nonsmokers decreased significantly from 0.065 ng/mL (95% CI [0.052, 0.081]) to 0.029 ng/mL (95% CI [0.026, 0.033]) (Table 1), representing an overall decrease of 55.0% (*p* < 0.0001). Serum COT concentrations among nonsmokers fell by an average of 11.0% (95% CI [8.8%, 13.1%]; *p* < 0.0001) every 2 year cycle between 2003 and 2018 (Appendix A Table A4).

The GMs and 95% CIs for serum COT concentrations in U.S. nonsmokers by race and Hispanic origin from 2003 to 2018 are depicted in Figure 2 along with the log-linear regression lines for each racial and Hispanic origin group. Serum COT concentrations decreased significantly in all racial and Hispanic origin groups studied throughout this period (*p* ≤ 0.0013). Serum COT GMs of non-Hispanic Blacks dropped by 49.8% (*p* = 0.0013) while those of Mexican Americans fell by 44.5% (*p* = 0.0001), and those of non-Hispanic Whites decreased by 56.1% (*p* < 0.0001). From 2003 to 2018, the rate of decline of serum COT concentrations for Non-Hispanic Whites was 11.8% per NHANES survey cycle. Non-Hispanic Blacks had significantly higher (*p* < 0.0001) serum COT concentrations initially, but the rate of decline for Non-Hispanic Blacks was not significantly different from Non-Hispanic Whites. Similarly, over the same period, while Mexican Americans had lower initial serum COT concentrations compared to Non-Hispanic Whites, the rate of decline of serum COT concentrations was not significantly different than that of Non-Hispanic Whites (Appendix A Table A5). In the final study period, pairwise comparisons show the serum COT concentrations of the non-Hispanic Black population are 150% higher than those of non-Hispanic Whites and over 200% higher than those of Mexican Americans (*p* < 0.0001) (Table 2).

The GMs and 95% CIs for serum COT concentrations in U.S. nonsmokers by age group from 2003 to 2018 are depicted in Figure 3 along with the log-linear regression lines for each age group. COT concentrations for all age groups showed significant decreases over time (*p* values ≤ 0.022). The largest decrease in exposure was seen in children aged 6–11 years, whose GMs fell from 0.123 ng/mL (95% CI [0.082, 0.186]) to 0.043 ng/mL (95% CI [0.036, 0.051]) (Table 1), representing more than a 65% reduction in GM serum COT concentration. The rate of decline for adults aged 20–39 years was 9.1% per NHANES survey cycle (*p* < 0.0001). Based on our regression model, age groups below adults aged 20–39 years had significantly higher (*p* < 0.0001) initial serum COT concentrations than adults aged 20–39 years. U.S. nonsmokers aged 60–85 years had significantly lower (*p* = 0.0003) initial serum COT concentrations than adults aged 20–39 years. The rates of decline in serum COT concentrations for both children aged 6–11 years and adolescents aged 12–19 years were faster (*p* ≤ 0.0452) than adults aged 20–39 years during this period. The rates of decline for other age groups were not significantly different than that of adults aged 20–39 years (Appendix A Table A6). While children saw decreases in their exposure throughout the study period, younger populations remain more exposed, as measured by serum COT GM concentrations, than older populations. Notably, despite a reduction of more than 55% (*p* < 0.0001) in their exposure, during the final study period (2017–2018), children aged 3–5 years were significantly more exposed when compared to all other age groups in the study (*p* ≤ 0.0002) (Table 2).

The GMs and 95% CIs for serum COT concentrations in U.S. nonsmokers by sex from 2003 to 2018 are depicted in Figure 4 along with the log-linear regression lines for each sex. COT concentrations for both the male and female populations dropped significantly during this survey period, by 59.1% (*p* < 0.0001) and 51.3% (*p* < 0.0001), respectively. The rate of decline in serum COT concentrations for females was 9.8% per NHANES survey cycle. While males had significantly higher (*p* < 0.0001) concentrations of serum COT initially, their concentrations declined faster (*p* = 0.0002) than females during this period (Appendix A Table A7). Despite a decrease in the disparity between their exposures over time, a pairwise comparison of these groups during the most recent survey period (2017–2018) showed that the differences between males and females remained significant, with males having approximately 13.2% (*p* = 0.0384) higher serum COT concentrations than females (Table 2).

The long-term precision of the NHANES serum COT method is displayed in Figure 5. Despite changes to the method, and storage for over 20 years, all consistency measurements fell within the 95% CIs of the characterization means established for all three QC pools during NHANES III. Additionally, the slopes of each of the three QC pools over time were not significantly different from zero (*p* ≥ 0.1829).

## 4. Discussion

In 2006, Pirkle et al. reported a decline of approximately 70% in the GM of serum COT concentrations in U.S. nonsmokers from 1988 to 2002 [16]. An evaluation of prevalence in 2018 examined findings from NHANES from its outset to 2014 and suggested that progress appeared to have stalled in reducing the prevalence of SHS exposure among nonsmokers [22]. A later analysis of SHS prevalence by the same group covering the period 2011–2018 and using the current LOD of 0.015 ng/mL concluded that, in fact, the prevalence of SHS exposure among nonsmokers was continuing to decline [23]. Our analysis uses the GM of continuous serum COT concentrations from the last 16 years of NHANES data to describe how the magnitude of SHS exposure of the U.S. nonsmoker population is changing over time and provides clear evidence that a decline in mean exposure to SHS, generally and across groups by race and Hispanic origin, age, and sex, has continued since 2002. Our findings are in agreement with the conclusions of Tsai et al. [23], based on the prevalence of SHS exposure, and are likely a result of decreased prevalence of smoking, increased awareness of the risks for SHS exposure, and the adoption of comprehensive smoke-free laws prohibiting smoking in U.S. workplaces and public spaces which has progressed during this time [2,22,24].

For this analysis, we used a 10 ng/mL serum COT cutoff point for defining nonsmokers. While many researchers have attempted to establish an optimal serum COT concentration for distinguishing people who smoke from nonsmokers, there continues to be a lack of standardization among studies [25]. Univariate and regression analyses, completed for analysis of NHANES III data from the period 1988–1991, demonstrated minimal difference in results whether using 10 or 15 ng/mL as the serum COT cutoff [15]. However, as the magnitude of SHS exposure declines, the optimal serum COT cutoff point for distinguishing smokers from nonsmokers should also decline. While 10 ng/mL remains a commonly used cutoff, an evaluation of NHANES data using receiver operating characteristic curve analysis proposed several different cutoffs based on different demographic groupings as well as a new overall cutoff point of 3 ng/mL for minimizing the rate of misclassification [26]. Nonetheless, the 10 ng/mL cutoff continues to fall within the appropriate range recommended for use in serum COT assays that wish to distinguish those who smoke from nonsmokers [27]. Therefore, for consistency, we have continued to use the 10 ng/mL cutoff for defining the nonsmoker population across NHANES surveys.

Significantly higher serum COT concentrations of non-Hispanic Blacks as compared to other racial and Hispanic origin groups has concerned researchers for decades. However, this apparent disparity may be explained at least in part by genetic differences amongst a portion of the non-Hispanic Black population rather than differing exposures. UGT2B10 is the catalyst of COT glucuronidation. A significant percentage of African Americans do not exhibit UGT2B10 enzyme activity. Recent research found that higher serum COT concentrations in non-Hispanic Black people who smoke, as compared to other racial groups, result from lower levels of UGT2B10-catalyzed COT glucuronidation by a subset of African Americans [28]. Future research is required to confirm this phenomenon in nonsmoker populations. Analysis of free versus total urinary-COT biomarker data would allow for the exclusion of individuals null for UGT2B10 activity from the NHANES data set, perhaps revealing a more accurate assessment of exposure by race.

Our findings of decreased SHS exposure as measured by GM of serum COT concentrations among all age groups, including children, is consistent with a recent report from Tatten-Birch and Jarvis that showed a 90% decrease in SHS exposure among children aged 4–15 years in England from 1998 to 2018 [29]. Disparities in SHS exposure between children and adults have been noted previously and persist [16]. This may be partially explained by research showing the home is the primary source of exposure for children, and nearly all nonsmokers who live with a smoker are exposed to SHS [30]. Furthermore, children may be less aware of the health consequences of SHS exposure and less able to distance themselves from a smoker in their home. Encouraging voluntary smoke free rules in the home and inside vehicles as well as promoting cessation in adult smokers have been identified as potential interventions to reduce SHS exposure and its related health hazards in children [31,32].

Our study has many strengths. It takes full advantage of the complex survey design of NHANES, allowing the data in this report to accurately measure exposure across the U.S. population generally and within a number of different demographic groups. Our study is multifaceted. We examined changes in serum COT concentrations within and across demographic groups both in absolute and relative terms. We also studied trends in serum COT concentrations during this period, overall and among several demographic groups. For comparisons over time, it is critical that changes to the laboratory procedure do not affect analytical measurements. The long-term precision data confirm no systematic drift has occurred in our measurements of serum COT concentrations during the more than 25 years that the assay has been active, providing additional assurance for anyone who relies on the NHANES serum COT data for their research.

Our study also has a few limitations. We relied on self-reported questionnaire data as part of the nonsmoker definition; however, the use of serum COT as another measure of nonsmoker status helps mitigate this limitation. Additionally, declining serum COT concentrations have decreased detection rates and created a challenge for accurate estimation of GMs (Appendix A Table A2). Finally, exposure to exhaled emissions from e-cigarettes and other non-combustible, inhaled nicotine devices may slightly increase serum COT concentrations in nonusers. While e-cigarettes eliminate exposure from sidestream smoke, researchers have measured traces of airborne nicotine and other toxicants in an exposure chamber following e-cigarette use [33]. People using their typical e-cigarette brand tend to absorb nicotine efficiently such that minimal nicotine is exhaled [14]. These trace levels of exhaled nicotine can be inhaled by nonusers and result in measurable secondhand exposure in vape conventions and other social settings [34,35,36]. These exposures, while measurable, are substantially less than those produced by SHS exposure to cigarette smoke [37,38]. As use of noncombustible tobacco products increases, concurrent measurement of serum COT and additional biomarkers of exposure [39,40,41] may clarify assessments of exposure and risk.

## 5. Conclusions

From 2003 to 2018, mean serum COT concentrations in U.S. nonsmokers declined by 55.0%, from 0.065 to 0.029 ng/mL. While children aged 3–5 years and the non-Hispanic Black population continued to have relatively higher concentrations of serum COT than older populations and other racial groups, respectively, significant decreases in serum COT concentration were observed across all demographic groups regardless of age, sex, or race and Hispanic origin during this period. Long-term precision data amassed from measurements of QC pools over more than 25 years continue to reveal no systematic drift and demonstrate the long-term stability of the serum COT assay. Although there is no safe level of exposure to SHS, the continued decrease in serum COT concentrations found over this period is a positive public health outcome and demonstrates the importance and efficacy of adopting comprehensive smoke-free laws, policies, and rules for workplaces, public places, homes, and vehicles to fully protect nonsmokers from SHS exposure.

## Figures and Tables

**Figure 1 ijerph-19-05862-f001:**
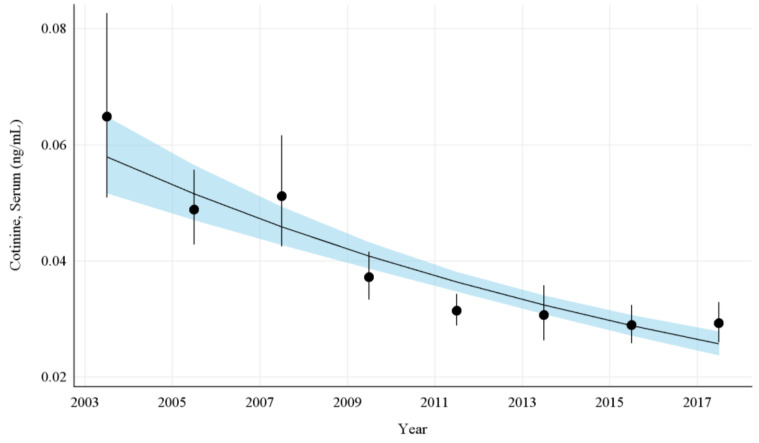
Sample-weighted geometric means (GM) and 95% confidence intervals (CI) of serum cotinine for all U.S. nonsmokers aged ≥ 3 years from 2003 to 2018. The data are plotted at the approximate midpoint of each study interval for eight separate NHANES survey cycles. The points represent the GMs and the vertical lines represent 95% CIs of the GMs for each survey cycle. The black trend line represents the weighted log-linear regression line and the shaded area is the 95% CI of the regression line.

**Figure 2 ijerph-19-05862-f002:**
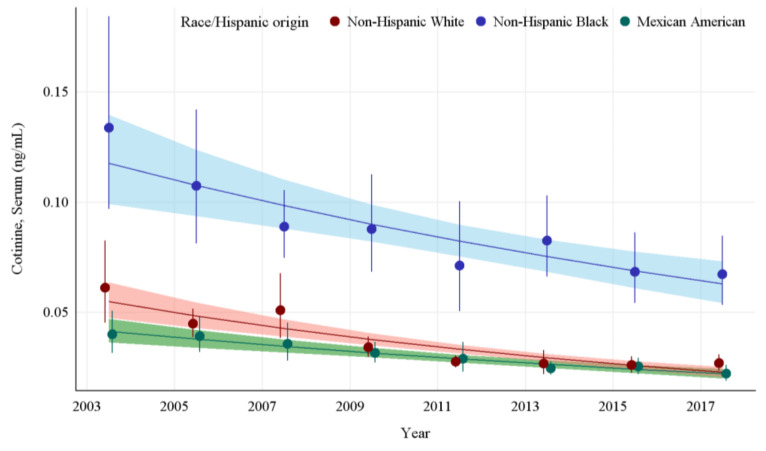
Sample-weighted geometric means (GM) and 95% confidence intervals (CI) of serum cotinine for U.S. nonsmokers aged ≥ 3 years by race and Hispanic origin from 2003 to 2018. The data are plotted at the approximate midpoint of each study interval for eight separate NHANES survey cycles. The points represent the GMs and the vertical lines represent 95% CIs of the GMs for each survey cycle. Data points from different racial groups within the same NHANES survey have been slightly offset to more easily visualize overlapping 95% CIs. The dark trend lines represent the weighted log-linear regression lines for each racial group and the corresponding shaded areas are the 95% CIs of the regression lines.

**Figure 3 ijerph-19-05862-f003:**
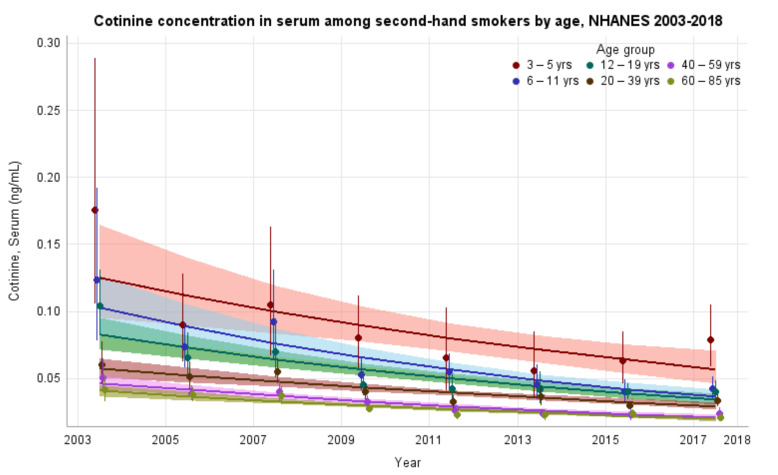
Sample-weighted geometric means (GM) and 95% confidence intervals (CI) of serum cotinine for U.S. nonsmokers aged ≥ 3 years by age group from 2003 to 2018. The data are plotted at the approximate midpoint of each study interval for eight separate NHANES survey cycles. The points represent the GMs and the vertical lines represent 95% CIs of the GMs for each survey cycle. Data points from different age groups within the same NHANES survey have been slightly offset to more easily visualize overlapping 95% CIs. The dark trend lines represent the weighted log-linear regression lines for each age group and the corresponding shaded areas are the 95% CIs of the regression lines.

**Figure 4 ijerph-19-05862-f004:**
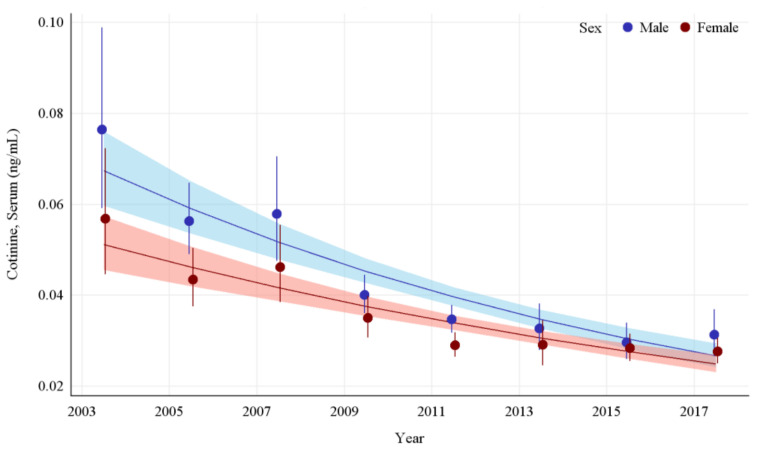
Sample-weighted geometric means (GM) and 95% confidence intervals (CI) of serum cotinine for U.S. nonsmokers aged ≥ 3 years by sex from 2003 to 2018. The data are plotted at the approximate midpoint of each study interval for eight separate NHANES survey cycles. The points represent the GMs and the vertical lines represent 95% CIs of the GMs for each survey cycle. Data points for each group within the same NHANES survey have been slightly offset to more easily visualize overlapping 95% CIs. The dark trend lines represent the weighted log-linear regression lines for each sex and the corresponding shaded areas are the 95% CIs of the regression lines.

**Figure 5 ijerph-19-05862-f005:**
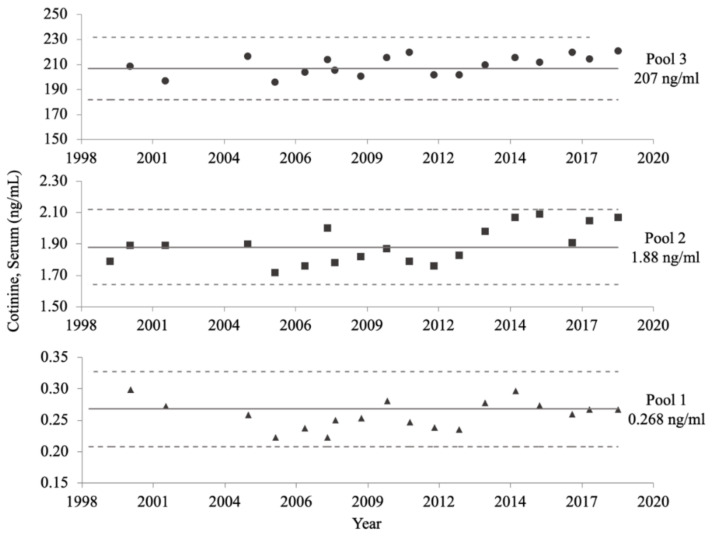
NHANES serum cotinine (COT) method long-term precision data. The mean COT concentration of each quality control (QC) pool, calculated in 1992, is represented by a solid line. The upper and lower limits of the 95% confidence intervals of the characterized means for each QC pool, also calculated in 1992, are denoted by dotted lines. Subsequent measurements taken from 1999 to 2018 are represented by solid circles, squares, and triangles, for the high, medium, and low concentration QC pools, respectively.

**Table 1 ijerph-19-05862-t001:** Sample-weighted geometric means (GMs) [and 95% confidence intervals (CIs)] for serum cotinine concentrations in U.S. nonsmoker populations from 2003 to 2018. The number of study participants is also listed by group and survey period (n).

Category	2003–2004	2005–2006	2007–2008	2009–2010	2011–2012	2013–2014	2015–2016	2017–2018
Overall	0.065 [0.052, 0.081] n = 6130	0.049 [0.043, 0.055] n = 6114	0.051 [0.043, 0.061] n = 5998	0.037 [0.034, 0.041] n = 6450	0.031 [0.029, 0.034] * n = 5938	0.031 [0.027, 0.035] * n = 6394	0.029 [0.026, 0.032] * n = 6207	0.029 [0.026, 0.033] * n = 5724
3–5 years	0.175 [0.110, 0.279] n = 420	0.090 [0.065, 0.125] n = 396	0.105 [0.070, 0.158] n = 364	0.080 [0.059, 0.109] n = 386	0.066 [0.043, 0.100] n = 360	0.056 [0.038, 0.082] n = 400	0.063 [0.048, 0.083] n = 394	0.079 [0.060, 0.103] n = 330
6–11 years	0.123 [0.082, 0.186] n = 832	0.073 [0.060, 0.090] n = 900	0.092 [0.066, 0.128] n = 973	0.053 [0.043, 0.066] n = 969	0.055 [0.044, 0.068] n = 988	0.046 [0.035, 0.059] * n = 1045	0.040 [0.034, 0.048] n = 994	0.043 [0.036, 0.051] n = 806
12–19 years	0.104 [0.084, 0.129] n = 1723	0.065 [0.052, 0.082] n = 1610	0.070 [0.052, 0.092] n = 868	0.046 [0.038, 0.054] n = 962	0.043 [0.035, 0.052] n = 943	0.042 [0.032, 0.054] * n = 1071	0.040 [0.036, 0.046] n = 998	0.040 [0.033, 0.048] * n = 899
20–39 years	0.060 [0.048, 0.076] n = 970	0.051 [0.043, 0.062] n = 1145	0.055 [0.047, 0.063] n = 1045	0.040 [0.036, 0.045] n = 1224	0.033 [0.029, 0.037] n = 1190	0.037 [0.031, 0.044] n = 1175	0.030 [0.027, 0.033] * n = 1178	0.034 [0.030, 0.038] n = 991
40–59 years	0.051 [0.041, 0.064] n = 828	0.039 [0.034, 0.044] n = 931	0.040 [0.034, 0.048] n = 1150	0.032 [0.029, 0.037] n = 1325	0.026 [0.024, 0.028] * n = 1180	0.024 [0.021, 0.027] * n = 1283	0.023 [0.020, 0.027] * n = 1242	0.024 [0.020, 0.028] * n = 1135
60–85 years	0.042 [0.034, 0.051] n = 1357	0.038 [0.034, 0.043] n = 1132	0.037 [0.032, 0.043] n = 1598	0.028 [0.025, 0.030] n = 1584	0.023 [0.020, 0.026] * n = 1277	0.023 [0.020, 0.026] * n = 1420	0.024 [0.021, 0.027] * n = 1401	0.021 [0.018, 0.024] * n = 1563
Male	0.076 [0.060, 0.097] n = 2822	0.056 [0.049, 0.064] n = 2775	0.058 [0.048, 0.070] n = 2824	0.040 [0.036, 0.044] n = 3041	0.035 [0.032, 0.038] n = 2781	0.033 [0.028, 0.038] * n = 2965	0.030 [0.026, 0.034] * n = 2854	0.031 [0.027, 0.036] * n = 2612
Female	0.057 [0.045, 0.071] n = 3308	0.043 [.038, 0.050] n = 3339	0.046 [0.039, 0.055] n = 3174	0.035 [0.031, 0.040] n = 3409	0.029 [0.027, 0.032] * n = 3157	0.029 [0.025, 0.034] * n = 3429	0.028 [0.026, 0.031] * n = 3353	0.028 [0.025, 0.030] * n = 3112
Non-Hispanic White	0.061 [0.046, 0.081] n = 2416	0.045 [0.039, 0.051] n = 2329	0.051 [0.039, 0.066] n = 2414	0.034 [0.030, 0.038] n = 2674	0.027 [0.025, 0.030] * n = 1781	0.027 [0.022, 0.032] * n = 2245	0.026 [0.023, 0.030] * n = 1808	0.027 [0.024, 0.031] * n = 1820
Non-Hispanic Black	0.134 [0.099, 0.180] n = 1622	0.107 [0.083, 0.139] n = 1593	0.089 [0.075, 0.104] n = 1229	0.088 [0.069, 0.111] n = 1101	0.071 [0.052, 0.098] n = 1550	0.082 [0.067, 0.101] n = 1279	0.068 [0.055, 0.085] n = 1209	0.067 [0.054, 0.083] n = 1183
Mexican American	0.040 [0.032, 0.050] n = 1617	0.039 [0.032, 0.047] n = 1704	0.036 [0.028, 0.044] n = 1340	0.032 [0.027, 0.036] n = 1532	0.029 [0.023, 0.036] n = 850	0.025 [0.022, 0.028] * n = 1219	0.025 [0.022, 0.029] * n = 1323	0.022 [0.019, 0.026] * n = 952
Other Hispanic/Other Race/Multi-Racial	0.066 [0.055, 0.079] n = 475	0.045 [0.037, 0.053] n = 488	0.044 [0.036, 0.054] n = 1015	0.034 [0.029, 0.039] n = 1143	0.032 [0.027, 0.036] n = 1757	0.033 [0.028, 0.038] n = 1651	0.028 [0.024, 0.031] * n = 1867	0.029 [0.026, 0.033] * n = 1769

* GMs and 95% CIs were calculated with weighted detection rates below 60%.

**Table 2 ijerph-19-05862-t002:** Pairwise comparisons of sample-weighted serum cotinine geometric means (GM) between different demographic subgroups during the 2017–2018 NHANES survey period.

Category 1	Category 2	Log Difference	Standard Error	Adjusted *p*-Value	Percent Difference *
3–5 years	6–11 years	0.266	0.059	0.0002	−45.8
3–5 years	12–19 years	0.296	0.047	<0.0001	−49.4
3–5 years	20–39 years	0.370	0.051	<0.0001	−57.4
3–5 years	40–59 years	0.523	0.059	<0.0001	−70.0
3–5 years	60–85 years	0.578	0.052	<0.0001	−73.6
6–11 years	12–19 years	0.030	0.034	1.0000	−6.67
6–11 years	20–39 years	0.104	0.045	0.3409	−21.3
6–11 years	40–59 years	0.257	0.050	<0.0001	−44.7
6–11 years	60–85 years	0.312	0.032	<0.0001	−51.2
12–19 years	20–39 years	0.074	0.039	0.9295	−15.7
12–19 years	40–59 years	0.227	0.046	<0.0001	−40.7
12–19 years	60–85 years	0.282	0.032	<0.0001	−47.7
20–39 years	40–59 years	0.153	0.035	0.0005	−29.7
20–39 years	60–85 years	0.208	0.034	<0.0001	−38.0
40–59 years	60–85 years	0.055	0.039	1.0000	−11.8
Female	Male	−0.054	0.026	0.0384	13.2
Non-Hispanic White	Non-Hispanic Black	−0.398	0.041	<0.0001	150.0
Non-Hispanic White	Mexican American	0.084	0.031	0.0491	−17.6
Non-Hispanic White	Other Hispanic/Other Race/Multi-Racial	−0.034	0.035	1.0000	8.2
Mexican American	Non-Hispanic Black	−0.482	0.041	<0.0001	203.5
Mexican American	Other Hispanic/Other Race/Multi-Racial	−0.119	0.038	0.0153	31.3
Non-Hispanic Black	Other Hispanic/Other Race/Multi-Racial	0.364	0.038	<0.0001	−56.7

* Percent difference was calculated as (GM Category 2 − GM Category 1)/GM Category 1 * 100 (%) = (1/10^log difference − 1) * 100, log difference = log10 GM Category 1 − log10 GM Category 2.

## Data Availability

All NHANES data are publicly available for download at https://www.cdc.gov/nchs/nhanes/index.htm (accessed on 2 February 2022).

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
