# Peer review of "Geometric Mean Serum Cotinine Concentrations Confirm a Continued Decline in Secondhand Smoke Exposure among U.S. Nonsmokers—NHANES 2003 to 2018"

_ijerph, 2022, doi:10.3390/ijerph19105862_

Round 1

Reviewer 1 Report

Caron and coworkers quantified cotinine concentrations in NHANES serum 19 COT results from 8 continuous NHANES 2-year cycles from 2003 to 2018 using LC-MS/MS. The results showed that from 2003 to 2018, mean serum COT concentrations in U.S. nonsmokers decreased by 55.0%, from 0.065 ng/ml to 0.029 ng/ml. Disparities were observed among different age groups and demographic groups. Both Children aged 3-5 years and the non-Hispanic Black population were found to have relatively higher concentrations of serum COT than older population sand other racial groups. Overall, the manuscript was well written and can be accepted after some minor revisions outlined below.

  1. The author reported a LOD (limit of detection) of 0.015 ng/mL, what about LOQ (limit of quantification)? Or did the author mean LOQ when they said LOD?

  1. The extraction procedure was modified to include the recovery of hydroxy-cotinine. However, no hydroxy-cotinine results were reported.

  1. The data show that the COT level decreases as age increases. For example, Table 1, shows that COT levels order are: 3-5 years > 6-11 years > 12-19 years >20- 39 years…. While the author suggested that younger children be exposed more to secondhand smoke (SHS), it may actually suggest something different, ie, ability to metabolite COT to other metabolites changes as age increases.

  1. Calibration curve was missing. So were other method validation results such as, linearity, LOD and LOQ.

Author Response

Dear Reviewer 1,

Thank you for your thoughtful review of our manuscript. Below, please find a point-by-point response to each of your comments:

Point 1: The author reported a LOD (limit of detection) of 0.015 ng/mL, what about LOQ (limit of quantification)? Or did the author mean LOQ when they said LOD?

Response 1: The CDC’s Division of Laboratory Sciences follows CLSI recommendations by including both Type I and Type II error in defining the method LOD. Thus, this manuscript refers to LODs that are based on characterizing the relationship between the standard deviation of the measurement and concentration of the analyte at low concentrations. For this reason, we report numerical results above this carefully defined limit of detection and do not use the term “LOQ” in the data reports from our laboratory.

Point 2: The extraction procedure was modified to include the recovery of hydroxy-cotinine. However, no hydroxy-cotinine results were reported.

Response 2: This analysis used NHANES serum cotinine (COT) data from 2003 to 2018 to look at nonsmoker exposure to secondhand smoke (SHS). Hydroxycotinine (HCT) was added to the laboratory assay in 2013, thus we do not have as extensive a dataset to analyze. In addition, because HCT concentrations are much lower than COT concentrations in serum, they do not give us additional information on SHS exposure. In order to discuss all changes to the serum COT analysis throughout its history, we chose to note the addition of HCT as the reason for adding isopropanol to the eluting solvent during extraction.

Point 3: The data show that the COT level decreases as age increases. For example, Table 1, shows that COT levels order are: 3-5 years > 6-11 years > 12-19 years >20- 39 years…. While the author suggested that younger children be exposed more to secondhand smoke (SHS), it may actually suggest something different, i.e., ability to metabolite COT to other metabolites changes as age increases.

Response 3: As the reviewer suggests, we agree there are additional factors that may contribute to increased concentrations of serum COT amongst younger nonsmokers as compared to older nonsmokers. While differences in metabolism of nicotine have been noted due to genetic variation, studies of these differences due to age have been less conclusive. A 2006 report by Johnstone et al. acknowledged that their finding of a marginally increased nicotine metabolite ratio with increased age1 contradicted previous work that found nicotine metabolism was faster in younger individuals2. Nonetheless, much of this analysis was done in adult smoking populations. More study is needed to determine the extent to which nicotine metabolism affects serum COT concentrations in children as well as among nonsmokers.  Therefore, we chose to focus our discussion on possible explanations of this disparity that currently have the most supporting evidence in the literature.

Point 4: Calibration curve was missing. So were other method validation results such as, linearity, LOD and LOQ.

Response 4: The objective of this manuscript was to determine the long-term trends in serum COT concentrations as a measure of SHS exposure and to support the analysis with data and documentation of long-term assay stability. Method validation results including detailed discussion of calibration curve, linearity, as well as our method for determining the assay’s limit of detection are available in previous method manuscripts3 as well as the method documentation that can be found on the NHANES website https://wwwn.cdc.gov/nchs/nhanes/Default.aspx.

References

  1. Johnstone E., Benowitz N., Cargill A., Jacob R., Hinks L., Day I. et al. Determinants of the rate of nicotine metabolism and effects on smoking behavior. Clin Pharmacol Ther 2006; 80: 319– 330.
  2. Molander L, Hansson A, Lunell E. Pharmacokinetics of nicotine in healthy elderly people. Clin Pharmacol Ther. 2001;69(1):57-65. doi:10.1067/mcp.2001.113181
  3. Bernert, J.T.; Turner, W.E.; Pirkle, J.L.; Sosnoff, C.S.; Akins, J.R.; Waldrep, M.K.; Ann, Q.; Covey, T.R.; Whitfield, W.E.; Gunter, E.W.; et al. Development and Validation of Sensitive Method for Determination of Serum Cotinine in Smokers and Nonsmokers by Liquid Chromatography/Atmospheric Pressure Ionization Tandem Mass Spectrometry. Clin. Chem. 1997, 43, 2281–2291.

Reviewer 2 Report

This article is aimed to demonstrate the long-term trends of serum cotinine concentration in US non-smokers from 2003 - 2018 based on data from the National Health and Nutrition examination Surveys. However, the conclusions from this study are closely similar with those published by the same research group in 2021 (Tsai J, Homa DM, Neff LJ, Sosnoff CS, Wang L, Blount BC, Melstrom PC, King BA. Trends in Secondhand Smoke Exposure, 2011-2018: Impact and Implications of Expanding Serum Cotinine Range. Am J Prev Med. 2021 Sep;61(3):e109-e117.): â‘ Population-level secondhand smoke exposure is reducing;②Secondhand smoke exposure is higher among Children;③Non-hispanic Blacks has a higher prevalence of secondhand smoke exposure than other racial groups. It is recommended to take a more innovative perspective to analyze the data.

Author Response

Dear Reviewer 2,

Thank you for your thoughtful review of our manuscript. Below, please find a our response to your comments:

Point 1: This article is aimed to demonstrate the long-term trends of serum cotinine concentration in US non-smokers from 2003 - 2018 based on data from the National Health and Nutrition examination Surveys. However, the conclusions from this study are closely similar with those published by the same research group in 2021 (Tsai J, Homa DM, Neff LJ, Sosnoff CS, Wang L, Blount BC, Melstrom PC, King BA. Trends in Secondhand Smoke Exposure, 2011-2018: Impact and Implications of Expanding Serum Cotinine Range. Am J Prev Med. 2021 Sep;61(3):e109-e117.): ①Population-level secondhand smoke exposure is reducing;②Secondhand smoke exposure is higher among Children;③Non-hispanic Blacks has a higher prevalence of secondhand smoke exposure than other racial groups. It is recommended to take a more innovative perspective to analyze the data.

Response 1: The reviewer is correct that we reached several of the same conclusions as Tsai et al. after conducting analyses on an expanded range of the same dataset.  However, while we agree these two analyses are similar, the major difference between the previous publication and ours is that this manuscript takes a different approach to reporting and understanding secondhand smoke exposure (SHS) in U.S. nonsmokers. The previous analysis by Tsai et al. defines nonsmokers categorically as exposed or unexposed based on a serum cotinine (COT) concentration range of 0.015 – 10 ng/mL. Conversely, we are using the geometric mean of continuous serum COT concentrations to describe how the magnitude of SHS exposure of the U.S. nonsmoker population is changing over time. These estimates of serum COT concentrations among the study population have not been published in the past. We believe this is an important addition to the literature because the levels provide important information about background exposure among various populations and subgroups and can be used to determine how exposed any group is relative to the larger population of U.S. nonsmokers. Additionally, these values may be useful to researchers and public health professionals seeking to understand what base levels of exposure might be expected within different groups as a result of SHS exposure. Finally, our manuscript also includes long-term stability data and documentation on the history of serum COT analysis at CDC in order to provide additional support to our conclusions and assurance to other researchers who rely on the validity of the NHANES serum COT data set.

Reviewer 3 Report

The study aims to investigate long-term trends in serum cotinine concentrations, as a measure of secondhand smoke exposure, among U.S. nonsmokers using NHANES data from 2003 to 2018. Caron et al used chromatography-tandem mass spectrometry assay to analyze the serum cotinine results. Overall, they found the serum cotinine concentrations were declined by 55% in U.S. non-smokers from 2003 to 2018 whereas every 2 years concentrations were reduced by 11%. Serum cotinineconcentrations of non-Hispanic Black population, children between 3-5 years of age, men remained higher than other races, age groups and women respectively. The study was performed using large number of subjects which is good, and the manuscript is well written.

Few minor comments for the authors to consider:

  • Abstract Line 2: Add the full form of SHS.
  • Add more references in the introduction and discussion.

Author Response

Dear Reviewer 3,

Thank you for your thoughtful analysis of our manuscript. Below, please find a point-by-point response to each of your comments:

Point 1: Abstract Line 2: Add the full form of SHS.

Response 1: We have incorporated this suggestion and thank the reviewer for the feedback. We have also added the full spelling of the first use of “confidence interval” to our manuscript and made all uses of “ng/mL” consistent.

Point 2: Add more references in the introduction and discussion.

Response 2: We have reviewed our references in both the introduction and discussion sections and currently feel we have referenced the relevant literature for each.  However, we would be very open to adding additional relevant citations. We kindly ask that the reviewer to specify any statements that require additional support and would be very appreciative of any references they might suggest.

Round 2

Reviewer 2 Report

The author responed the review comments quite well, and I believe this study has certain value in the field of second-hand smoke research. I would still suggest that the title, structure and focus of the discussion of this article be revised to differentiate it from the previous article published earlier by the author's team.

Author Response

Dear Reviewer 2,

Thank you for your additional feedback on our manuscript. Below, please find a our response to your comments:

Point 1: The author responded the review comments quite well, and I believe this study has certain value in the field of second-hand smoke research. I would still suggest that the title, structure and focus of the discussion of this article be revised to differentiate it from the previous article published earlier by the author's team.

Response 1: We are grateful for the additional feedback from the reviewer and have decided to implement their suggestions. To differentiate this report from the previous article on prevalence of SHS exposure, we have made changes to the title of the manuscript as well as the introduction and discussion. We have also included additional language to differentiate measurements of SHS exposure using prevalence versus our use of geometric mean serum COT concentrations. All changes can be viewed in the version of our manuscript with tracked changes. We believe these changes address the concerns of the reviewer and improve the overall presentation of our analysis.